# Deep Learning Architecture to Improve Edge Accuracy of Auto-Contouring for Head and Neck Radiotherapy

**DOI:** 10.3390/diagnostics13132159

**Published:** 2023-06-24

**Authors:** Ryan Gifford, Sachin R. Jhawar, Samantha Krening

**Affiliations:** 1Department of Integrated Systems Engineering, The Ohio State University, 1971 Neil Ave, Columbus, OH 43210, USA; krening.2@osu.edu; 2Comprehensive Cancer Center, Department of Radiation Oncology, The Ohio State University, 410 W 10th Ave, Columbus, OH 43210, USA; sachin.jhawar@osumc.edu

**Keywords:** auto-segmentation, head and neck cancer, deep learning

## Abstract

Deep learning (DL) methods have shown great promise in auto-segmentation problems. However, for head and neck cancer, we show that DL methods fail at the axial edges of the gross tumor volume (GTV) where the segmentation is dependent on information closer to the center of the tumor. These failures may decrease trust and usage of proposed auto-contouring systems. To increase performance at the axial edges, we propose the spatially adjusted recurrent convolution U-Net (SARC U-Net). Our method uses convolutional recurrent neural networks and spatial transformer networks to push information from salient regions out to the axial edges. On average, our model increased the Sørensen–Dice coefficient (DSC) at the axial edges of the GTV by 11% inferiorly and 19.3% superiorly over a baseline 2D U-Net, which has no inherent way to capture information between adjacent slices. Over all slices, our proposed architecture achieved a DSC of 0.613, whereas a 3D and 2D U-Net achieved a DSC of 0.586 and 0.540, respectively. SARC U-Net can increase accuracy at the axial edges of GTV contours while also increasing accuracy over baseline models, creating a more robust contour.

## 1. Introduction

To create a radiotherapy treatment plan, radiation oncologists must draw “contours” on every slice of a patient’s treatment planning computed tomography (CT) scan to delineate where different doses of radiation should be directed and what organs at risk should be avoided (Figure 1). Manually contouring can take a clinician an average of 2.4 h per patient [1]. Time is a critical factor because delays decrease cure rates, decrease patient survival, and limit the number of patients oncologists can treat [2]. Manually contouring is also a repetitive, time-consuming process that does not make the best use of an oncologist’s expertise. Auto-contouring (AC) performed by software has been shown to reduce doctor-to-doctor variability in delineations of regions of interest [3,4], while also reducing the time it takes to produce a contoured region of interest [5]. The goals of our AC work are to make high-quality treatment plans more accessible to all patients, decrease time from diagnosis to treatment, and free oncologists from repetitive tasks.

Deep learning (DL) approaches to automate GTV contouring are the subject of active study [6]. While preliminary success has been achieved, the accuracy of the delineations decreases for smaller tumor sizes in head and neck cancer (HNC) [7,8].

In this paper we aim to answer three main research objectives.

The first research objective of this work is to prove that baseline AC algorithms for gross tumor volumes (GTV) of the head and neck create contouring errors at the top and bottom of all tumors. By examining these errors, we can begin to propose architectures aimed to address these weaknesses. This will show there are similar and systematic mistakes in the current auto-contouring approaches that need to be addressed to move the field forward.

The second research objective is to present a novel deep learning architecture that mirrors the clinician workflow, starting at the middle of the GTV and contouring the edges last. Saliency of the GTV decreases towards the edges. To help maintain its location, we both push and align what is important from the middle regions of the GTV, comparable to how clinicians look to the more salient portions of the tumor as they contour regions where it is harder to distinguish the GTV from normal tissue. To do this, we combine three concepts from computer vision to AC: spatial transformer networks (STN), convolutional recurrent neural network (CRNN) cells, and the U-Net architecture. Combining these components, we propose the spatially adjusted recurrent convolution U-Net (SARC U-Net). SARC U-Net pushes information from the middle of the GTV towards the top and bottom edges of the tumor to improve GTV accuracy, similar to the contouring workflow of radiation oncologists.

The third research objective of this work is to demonstrate that SARC U-Net will (1) increase the overall Sørensen–Dice coefficient (DSC) of proposed GTVs, and (2) increase the DSC at the top and bottom tumor edges (in the axial direction) of the GTV. Further, our goal is to increase accuracy with little addition to computational overhead, as models that take adjacent slices into context such as the 3D U-Net employ a high number of trainable parameters.

In this paper we propose SARC U-Net and a simpler variation that removes the STN component, the recurrent convolutions U-Net (RC U-Net). Both architectures are specifically aimed at increasing performance in the axial edges of the GTV that focus on inter-slice connectivity using recurrence between axial slices. Like the contouring workflow of radiation oncologists, we created an architecture that pushes information from the middle of the GTV towards the top and bottom edges of the tumor to improve GTV accuracy.

For this mature study, publicly available data was used to train and evaluate models from the Cancer Imaging Archive Head-Neck-Radiomics-HN1 dataset [9,10], described in Aerts et al. [11]. The full dataset comprises 137 HNC patients with GTV volumes contoured using CT and PET scans by an expert radiation oncologist.

### Related Work

Guo et al. [8] implemented a dense 3D U-Net to extract richer information from multi-modality information and increase accuracy. However, it was noted that performance drops dramatically for CT slices with a small tumor volume. Huang et al. [7] implemented a variation of the 2D U-Net and achieved a high DSC of 0.785. However, they removed all slices where the area of the tumor was less than 0.5 cm${}^{2}$. If we remove slices less than 0.5 cm${}^{2}$ in our dataset, 38% of patients would have slices removed from consideration. Of those slices, 87% are closer to the axial edges of the contour than to the center of the tumor, with 63% corresponding to the top-most or bottom-most edges. While Huang et al. [7] attributed removing these slices to the partial volume effect, we believe they are also inclined to poor performance due to being edge slices. While inter-observer variability in manual contouring GTV volumes has been studied, it often focuses on the variability of the full GTV volume rather than variability across individual slices. However, Nyholm et al. [12] found that uncertainty in the delineation of the prostate often occurred in the axial boundaries, and Zukauskaite et al. [13] found that most inter-observer variability occurred in HNC GTV delineation due to unclear boundaries when using MRI. While no studies were found investigating the inter-observer variability of the HNC GTV at the edge slices using CT-PET imaging, we believe that, clinically, these edge slices represent some of the more difficult areas to contour. Further, Lin et al. [5] explicitly notes poor performance at the cranial-caudal edges, with increased performance in the tumor midsections, validating the need for a model that can increase performance at the edges of the GTV. We will later show that this is the case for a wide implementation of DL-based models.

Creating an AC tool that cannot contour the top and bottom of a tumor accurately has real clinical impacts. Zabel et al. [14] found that Atlas-based AC methods had a lower average DSC compared with DL methods and required more re-contouring by clinicians, suggesting that a lower DSC indicates the need for manual adjustment. Clinicians may decide not to use the AC tool as it does not save much effort, given they will need to review the AC solution and manually contour the top and bottom of the contour. In addition, there is not a clear boundary on a single patient as to when the clinicians should or should not trust the AC tool, leading to an overall degradation of trust in the AC results.

The slices with small tumor areas that were removed from consideration in Huang et al. [7] often correspond to the edges of the full GTV and heavily depend on context from previous (closer to the middle) adjacent slices. While 3D variations of the U-Net [15] can take adjacent slices into context, they have been shown to have little to no improvement over the 2D U-Net in terms of overall accuracy for HNC [16].

Other methods that can take advantage of information between adjacent slices are long short-term memory (LSTM)-based architectures, which capture the spatial–temporal relationships from slice to slice. Some methods that have been proposed to take advantage of this type of architecture are the LSTM multi-modal U-Net (applied to brain tumor segmentation) [17] and Spider U-Net (applied to blood vessel segmentation) [18]. Xu et al. [17] added a convolutional LSTM layer after the decoder path of the U-Net to try to incorporate information between slices, while Lee et al. [18] added the convolutional LSTM at the bottom of the architecture.

These model architectures only use a convolutional LSTM layer in either the bottom of the U-Net or after the decoder path. As the LSTM-based U-Net provides benefits to the overall segmentation performance, we also implement a LSTM-based component; however we add it at multiple levels in the decoder portion of the U-Net to enforce inter-slice connectivity at different feature resolutions.

## 2. Materials and Methods

For this study, data from the Cancer Imaging Archive HN1 dataset [9,10] described in Aerts et al. [11] was used. The full dataset comprises 137 HNC patients who have undergone radiotherapy treatment. For each patient, a primary gross tumor volume segmentation is included.

For model training, patients without positron emission tomography (PET) scans or whose GTV spanned only two or fewer axial slices were excluded, leaving a total of 70 patients for model training and testing. CT images were sampled at a pixel spacing of [0.977 mm, 0.977 mm] with a slice thickness of 3 mm. PET images were sampled at a pixel spacing of [2.673 mm, 2.673 mm] and re-sampled to [0.977 mm, 0.977 mm] with a slice thickness of 3 mm. After resizing all slices of the CT scans, PET scans, and GTV masks to a resolution of 256 × 256, we also further cropped the full scans to the area where we would reasonably expect to see a GTV for HNC. This cropping was carried out through an aggregation of the positions of the GTVs across all 137 patients, leaving the final resolution for each patient scan at 64 × 64 × 64 in the x, y, and z dimensions, respectively. While this cropping is functional for this dataset, in the future this area may need to be expanded as the total number of patients is small and we could be cropping out relevant areas where a GTV for HNC could exist.

CT scans were clipped to values of −200 and 200 Hounsfield units and normalized 0 to 1 using min max normalization. Additionally, PET scans were normalized independently across patients to 0 and 1 using min max normalization.

For our analysis, 7-fold cross validation over the full 70 patients was performed.

For each patient, we manually selected the approximate center of the GTV, although this may be imperfect as tumors are often elongated and the center slice does not necessarily correspond to the most salient portion of the tumor. For each patient, we then fed in the superior and inferior halves of the tumor to the model: one from the center superiorly towards the skull base, and the other from the center inferiorly towards the thoracic inlet. Data were fed into the network this way so that the most salient information was at the beginning of the sequence. As the most salient information corresponding to the GTV is most often at the center slice of the GTV volume, we are pushing information from the center out to the top and bottom edge slices where the GTV depends heavily on previous slices.

Our novel architecture leverages spatial transformation networks [19] and CRNN to both transform and push relative information to the axial edges of the GTV. SARC U-Net is shown in Figure 2, and will be described throughout the rest of Section 3.

SARC U-Net works to improve auto-contouring accuracy for the top and bottom portions of each tumor. Figure 3 was made with the *HN1* dataset and shows that a large portion of CT slices contain a small tumor area at the axial edges of the GTV. However, many state-of-the-art AC methods ignore or are not able to achieve high accuracy for these sections of each tumor. A critical assumption of our method is that these sections are correlated with low accuracy not just because they are small, but because they are at the top and bottom edges of the full GTV volume and therefore depend more on previous slices as they lack the saliency of slices in the middle of the GTV. Therefore, our proposed model puts emphasis on inter-slice connectivity to increase accuracy in the edges. In the context of CT and PET scans, inter-slice connectivity can be defined as the connection of information over adjacent images, using the general direction of the spine as an axis.

To account for inter-slice connectivity, we used a CRNN cell after the upsampling operation and before the residual connection from the corresponding layer in the decoder, as seen in Figure 2. We opted for a CRNN over a convolutional LSTM or convolutional gated recurrent units (GRU) for computational efficiencies as convolution operations are expensive; an RNN cell is simpler in complexity than an LSTM or GRU. Further, given the natural growth patterns of tumors, it is preferable to weight information from directly previous slices higher than long-term information.

The recurrent block of our baseline CRNN consists of a 2D convolution with kernel size (3,3) followed by ReLU activation. The input is the previous hidden state and the current slice at spatial step *z*. The output of the recurrent block is then concatenated with the corresponding residual connection for its specific slice from the decoder layer, and an additional convolution is performed with kernel size (3,3) followed by ReLU activation. This can be seen in Figure 2 as the box after the upsampling operation and before the skip connection (without the spatial transformation step).

Figure 4 shows the displacement in the GTV when considering a window at the same position in the axial center of the GTV compared with an edge slice. In Figure 4, the center of the GTV has shifted 10 pixels to the right and 4 pixels down from the center to the edge. When calculating the displacement in the GTV from slice to slice across the z-axis, the GTV shifts on average 1.5 pixels on the x-axis and 1.53 pixels on the y-axis. While larger kernel sizes may be able to capture both centers across two or more slices, they will never be appropriately aligned, which could lead to erroneous results the farther we get from the center of the GTV. Pre-existing methods that take advantage of inter-slice connectivity, such as 3D CNNs or CRNNs, convolve or push information forward uniformly across the same pixel locations, even if those pixels are inadequately aligned.

We developed a novel method of accounting for small shifts in head and neck anatomy from slice to slice using spatial transformer networks to address this issue in AC.

In CT and PET scans, we can consider the displacement of an object from frame to frame (slice to slice on the axial plane) as its relative motion from end point to end point. Typically, methods such as optical flow could be employed to track a moving region of interest. Optical flow estimation, however, relies on consistent lighting conditions [20] that may not hold in such imaging, given: (1) the decreasing intensities of PET scans as we move towards the edges of the GTV; and (2) scatter from dental amalgam and other such anomalies in the CT scans.

We captured motion in a GTV by introducing a learned affine transformation of the hidden state in our CRNN blocks. This transformation is applied to the previous hidden state before applying the next state updates.

Spatial transformer networks [19] are trainable modules that allow networks to be more robust to spatial invariances such as rotations, translations, scalings, etc. In a CNN, they can allow a network to learn a 6-parameter affine transformation, which can be applied to the input using grid coordinates and bi-linear interpolation. Learned spatial transformations were originally applied to the distorted Modified National Institute of Standards and Technology database, where the model learned to transform the images so that was is more robust to rotated, shifted, and scaled images.

Instead of using spatial transformations to account for invariance, in our network we used the STN module to learn appropriate transformations of the hidden state in the proposed CRNN structure to account for inter-slice connectivity better. As the internal structures of neighboring slices are not perfectly aligned, by convolving over the corresponding feature maps between frames with axial depth, *d*, at a single frame pixel position (x,y) and kernel size, *k*, we may begin to push forward information from the previous slices to inadequate spatial locations in the next slice.

To account for this, we allowed for a scaling, translation, rotation, or cropping of the hidden state in accordance to the next frame before applying the typical CRNN operation. ((3,3) convolution with ReLU activation.) By employing these transformations, the goal is to allow a mechanism for the model to align the feature maps of the hidden state to the feature maps of the next frame to push forward information to the appropriate spatial locations. Going back to an earlier example, we applied a transformation to account for the shift in the GTV from one slice to another; this allows us to correctly align the GTV-related features as we move from the center of the GTV to the edge.

This module, while not explicitly optimized on a loss function to account for motion, allows for a transformation that can increase accuracy of the overall network. This is described in Figure 5, while the localization network is described in Figure 6.

We placed the SARC block (i.e., the combination of CRNN and STN components) directly after the transposed convolution but before the merging; this ensures that information is adequately localized across slices before we bring in the feature maps from the encoder block with the same spatial resolution. To learn the appropriate transformation, we passed forward the feature maps before the upsampling operation as they are a denser representation of the current feature space. These features were then concatenated with a downsampled hidden state (through average pooling) and passed through a 3 × 3 convolution with a ReLU activation function. Through a reshaping of the feature space, as described in Figure 6, the network looks independently at each pixel and, through a series of fully connected layers, and outputs a 6-parameter affine transformation for each pixel’s feature space learned by the prior convolution. We then take the global max across all pixels for all 6 parameters in the learned spatial transformation. This transformation is then applied to the previous hidden state before the next state updates.

The final architecture is depicted in Figure 2. To ensure that it is the addition of the SARC block that increases performance, we tested this architecture in four different ways: (1) with the addition of the SARC block (SARC U-NET); (2) with only the CRNN but no STN component (RC U-Net); (3) a 2D U-Net; and (4) a 3D U-Net. The 2D and 3D U-Nets lack the SARC block and CRNN components. The RC U-Net follows the same architecture as SARC U-Net but only uses the CRNN with no STN component. A detailed view of the RC block is shown in Figure 7. While the SARC, RC, and 3D U-Nets can capture a sense of inter-slice connectivity, the 2D U-Net would have no way to model inter-slice connectivity.

The number of parameters for each model is shown in Table 1. SARC U-Net has some additional overload compared with the RC U-Net, but has 22% less complexity than the 3D U-Net. Although RC U-Net and SARC U-Net both have less complexity than the 3D U-Net, they have longer training and inference times as they have to process information sequentially. In this study, no optimizations were carried out to try and improve the speed of this process. However, due to this recurrent operation, SARC U-Net also has the ability to localize information from prior spatial steps during its recurrent step. Additionally, while training times are higher for the SARC and RC U-Net, both trained models can contour a GTV for a full patient scan (CT and PET Scan) in under 1 s, so this is not a limiting factor in the clinician’s workflow. However, in clinical settings, these times are subject to change as we begin to train on larger patient populations and potentially higher CT and PET resolutions.

Table 2 shows the differences between our proposed architectures and prior related work. Comparatively, [8,17,18] also propose models that can process inter-slice connectivity. However, we also employ a data-feeding strategy, which begins contouring from the center of the GTV. When this strategy is employed, we can push information from the most salient regions of the ROI to the least salient regions (center to edges). While this may not be applicable to architectures not applied to HNC, for our task the edges of the GTV are harder to distinguish against soft tissue comparatively with the larger, center portions of the GTV. Additionally, SARC U-Net is the only architecture that employs spatial transformations to align anatomy in prior imaging slices before the recurrent step.

While our SARC U-Net architecture extends off a 2D U-Net, it can also fit into more recent U-Net-based architectures. Both TransU-Net [21] and U-Net++ [22] are 2D U-Net-based architectures used for segmentation. TransU-Net uses a vision transformer in the encoder portion of the U-Net, while U-Net++ uses nested, dense skip connections to connect the encoder and decoder portions of the U-Net more strongly. Each architecture has a decoder phase that employs an upsampling operation followed by a skip connection and a convolution. Therefore, as proposed in our SARC U-Net, we can plug in SARC blocks to both U-Net++ and TransU-Net. To test the SARC block in these architectures, we implemented TransU-Net and U-Net++ with (1) 2D convolutions (original architectures); (2) 3D convolutions; (3) a CRNN after each upsampling operation; and (4) SARC blocks after each upsampling operation.

To train each model, we used a 2D-based DSC loss, where DSC is computed over the z-axis independently. Interestingly, the use of a 3D DSC metric (computing the Dice over the full tumor volume) for loss could contribute to current algorithms ignoring the top and bottom edges of each tumor, as Figure 3 shows they are most often the smallest portions of the tumor. 2D Dice loss gives every slice equal weight when computing loss, whereas an unmodified volumetric-based Dice loss will naturally give a higher weight to the larger portions of the tumor. All models are trained with a Dice loss that is averaged over all slices (2D Dice loss) rather than the full volumes, although the 3D-based Dice score is also reported after cross-validation.

To allow our network to take advantage of information from both CT scans and PET scans, we passed them into the network together. Since both scans are in grey scale, they have one input channel. To leverage the multi-modal information, we stacked them together so that the input channels to our network is 2, making the final shape of the input data (32,64,64,2).

All models were built using PyTorch and trained for 100 epochs using the Adam optimizer with an initial learning rate of $3\times 10^{-4}$ and a batch size of 10. No data augmentation methods were used for this study. Models were trained on the Ohio Supercomputer using a single NVIDIA Volta V100 with 32 GB GPU memory.

## 3. Results

K-Fold cross validation (7 folds) was used to obtain an average DSC compared with ground truth, defined as the provided expert GTV contour, for both baseline models and our novel architecture. All patients used for training had an associated CT and PET scan.

We report average 2D DSC, 3D DSC, 2D sensitivity, and 2D specificity for all model tests. Results are reported in Table 3.

Paired *t*-tests were performed by collecting all testing folds and averaging DSC across model bases (U-Net, TransU-Net, and U-Net++), and individually for each U-Net architecture. Across all tested architectures, SARC U-Net shows a significant improvement (*p*-value < 0.05), with a DSC of 0.611 compared with a DSC of 0.586 for the 3D architectures, 0.540 for the 2D architectures, and 0.581 for the proposed RC U-Net architecture (which does not use the STN component). This is exciting as the SARC U-Net has 22% fewer parameters than the 3D U-Net. This also proves that the improvement in accuracy is not achieved by simply increasing the number of trainable parameters, but rather by applying mechanisms for the network to model the problem better. Further, our RC U-Net (the simplified version of the SARC U-Net) has roughly half the number of parameters as the 3D U-Net, but is not significantly different in performance from the 3D U-Net (*p*-value = 0.3994) variants. Equal or better performance can be achieved with fewer trainable parameters if the architecture is tailored to the problem.

For the GTV volumes, the U-Nets that use SARC blocks show a significant increase in performance, with a DSC of 0.572 compared with 0.551 for the 3D architectures, 0.553 for the 2D architectures, and 0.544 for the RC architectures. This is comparable to the results found in Andrearczyk et al. [16], where there were small differences in DSC between the 2D and 3D architectures. In part, this is due to how DSC is calculated over the full volume, giving a high accuracy if the large (more obvious) portions of the tumor are captured. While the 2D U-Net shows a decrement in performance when DSC is averaged over all slices, Figure 8 shows this is because of poor performance at the top and bottom edge slices, which have less weight when computing DSC over the volume because they also take up a smaller proportion of the total tumor volume, as shown in Figure 3. The SARC U-Net, however, is designed to improve performance at the edge slices and increases both the total volumetric DSC and 2D DSC.

The first research objective of this work is to show that baseline auto-contouring algorithms for gross tumor volumes (GTV) of the head and neck create contouring errors at the top and bottom of tumors. We have already shown in Figure 3 that, at the top and bottom of each tumor, there exists a significant number of CT slices that contain a small GTV area, which corresponds to low accuracy regions. Figure 8 and Figure 9 show how the accuracy of each model varies from the center of the GTV to the top and bottom edges. While all tested AC models performed worse at the edges, our novel approach specifically targets and improves accuracy at the top and bottom of tumors. Figure 8 shows that the SARC U-net has an extremely significant 19.3% increase (0.49 vs. 0.30) in DSC over the 2D U-net at the farthest superior edges, an 11% increase (0.50 vs. 0.39) at the farthest inferior edges, and similar performance at the center of the GTVs. Over the 3D U-Net, the SARC U-Net shows an increased DSC of 4% (0.50 vs. 0.46) at the farthest inferior edges and 2% (0.49 vs. 0.47) at the farthest superior edges. There are similar and systematic mistakes in the current auto-contouring approaches for HNC that need to be addressed to move the field forward; mainly we need to focus more on improving accuracy throughout all locations of a tumor, not just the middle slices.

The Hausdorff distance is another performance metric commonly used for segmentation performance. Figure 9 shows the Hausdorff distance by slice distance from the middle of GTV. When comparing Hausdorff distances, the SARC U-Net outperforms all architectures at all slice locations.

Figure 10 show the actual proposed contours for a single patient of the test set compared with the ‘ground truth’ prepared by a radiation oncologist.

There are issues regarding the baseline algorithms, especially toward the end of the tumor, that will hinder the usability and acceptance of the baselines in a clinical setting. In Figure 10, the 2D U-Net shows some recovery at the edges of this contour but has issues of both over- and under-contouring throughout the GTV. Errors in over-contouring have the potential for increased toxicity, while errors in under-contouring could result in a geographic miss and increased risk of locoregional failure. For the 3D U-Net, as the contours get closer to the edges the model loses track of the GTV completely. It has some of instances of over-contouring, but many errors of under-contouring. A clinician is less likely to trust or use an AC tool that is inconsistent or consistently inaccurate.

The case shown in Figure 10 is one that was challenging for baseline U-Nets to contour the edges adequately. However, there are many factors that can increase uncertainty in the boundary regions of the GTV, making the edges challenging to contour, including low contrast between the tumor and surrounding soft tissue, and poor activations in the PET scan. Because of this, even in SARC U-Net, the central part of the tumor is more accurate than the top and bottom edges; however, Figure 8 shows that SARC U-Net performs better than all other models on average at the top and bottom tumor edges. Figure 10 shows what this looks like in a case where SARC U-Net was able to maintain its performance when all other models failed to contour the edges adequately.

## 4. Discussion

K-Fold cross validation (7 folds) was used to obtain an average DSC, compared with ground truth, defined as the provided expert GTV contour, for 2D and 3D U-Nets and our novel architecture. The average DSC is reported for both 2D and 3D architectures as well as two versions of our architecture: RC U-Net and SARC U-Net. We also tested these architectures over two additional U-Net variants: TransU-Net and U-Net++. Results are reported in Table 3 for the average 2D DSC and for the average across volumes (3D DSC).

With an average DSC of 0.611, our SARC U-Net showed an improvement in DSC of 2.5% over the 3D architectures (average DSC 0.586), 7.1% over the 2D architectures (average DSC 0.540), and 3% over the proposed RC U-Net (average DSC 0.581). Other recent proposed methods that consider inter-slice connectivity between slices have not seen performance increases as significant, although not all mentioned architectures are tailored to HNC specifically. Guo et al. [8] proposed Dense-Net, which performed 2% better over a baseline 3D U-Net (0.71 vs. 0.69). Xu et al. [17] and Lee et al. [18] proposed U-Nets modified with an LSTM, and achieved increases of 1.3% in DSC (0.7309 vs. 0.7171) and 4.8% in DSC (0.793 vs. 0.745) compared with a 2D U-Net. However, our method saw a 7.1% increase (0.611 vs. 0.540) in performance over a baseline a 2D U-Net. The SARC U-Net can increase performance with substantially fewer parameters than the 3D U-Net. For our task, the novel SARC U-Net is significantly better at modeling inter-slice connectivity than the baseline architectures.

With fewer parameters, this also proves that higher accuracy is not achieved through increased complexity, but rather by applying carefully designed mechanisms for the network to model the problem at hand better.

The SARC U-Net generates contours with a more natural growth pattern from the center to the edges of the tumor, since the contour of each slice is a direct continuation of the slice that comes before it. In particular, the SARC U-Net contours are impressive, not only for their accuracy, but also because the contours progress in a much more logical way in terms of how they gradually change. This is much closer to how a contour would be drawn by a human. In future work, we will explore how much of an impact that has on usability and adoption of an AC tool by clinicians.

Additional techniques to improve model performance should also be explored in future studies. Employing data-augmentation techniques would be useful during the training phase of the SARC U-Net to help make the model more robust and artificially increase the size of our small dataset. For the purposes of this study, we focused on architectural changes to help improve performance, but those changes are still limited by the size of the dataset.

While DSC was used to train and evaluate our model performances, it may not be the most effective measurement when it comes to real-world applications. Without any additional oversight, all errors in over-contouring are seen as the same regardless of the distance from the primary GTV, as explored in Nikolov et al. [23]. Expanding upon this, in Figure 10, the 3D U-Net contours the spinal cord. While this is seen as an error, it is no different from if the model had contoured the vocal cord or the pharynx, each of which might have drastically different clinical impacts when it comes to over-contouring. Accurately delineating a tumor that needs a full radiation dose and avoiding organs at risk that should not be irradiated has much more impact in the clinical outcomes of a patient and is something that DSC does not account for. In future work, we hope to explore segmentation metrics that account for potential clinical impacts in terms of its own error.

## 5. Conclusions

Current baseline DL models fail to contour the axial edges of GTVs adequately for HNC. In this study, we proposed the SARC U-Net, a model that leverages CRNNs to push information from the most salient portions of the GTV out to the axial edges, similar to the clinician contouring workflow. By adding an STN component to account for shifts in the location of the GTV as we move out towards the edges, we can increase the performance in contouring the edges and outperform baseline DL models.

## Figures and Tables

**Figure 1 diagnostics-13-02159-f001:**
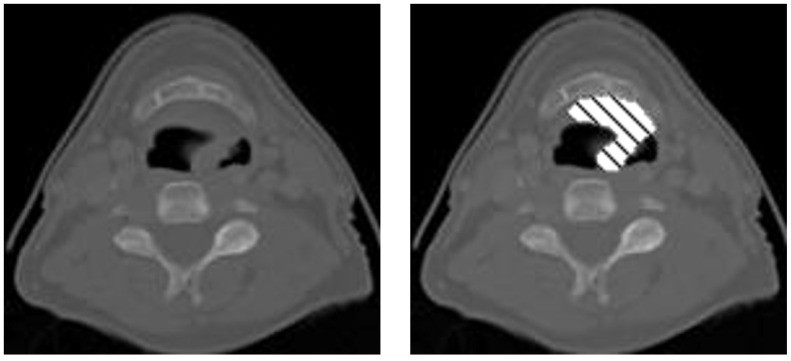
A CT slice of a patient’s head. The gray-scale portion is the CT image, and the cross-hatched pattern is the contour an oncologist drew on top of the CT scan to plan where radiation should be directed during treatment.

**Figure 2 diagnostics-13-02159-f002:**
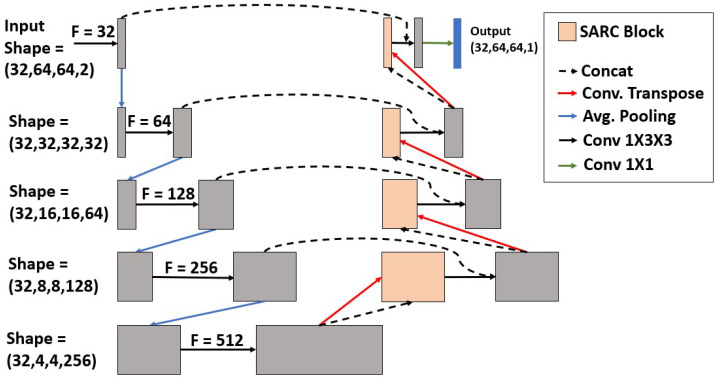
Overall architecture. U-net with SARC blocks. F Corresponds to the filter size for a given convolution operation, and the shape corresponds to (sequence length, height, width, and filter size).

**Figure 3 diagnostics-13-02159-f003:**
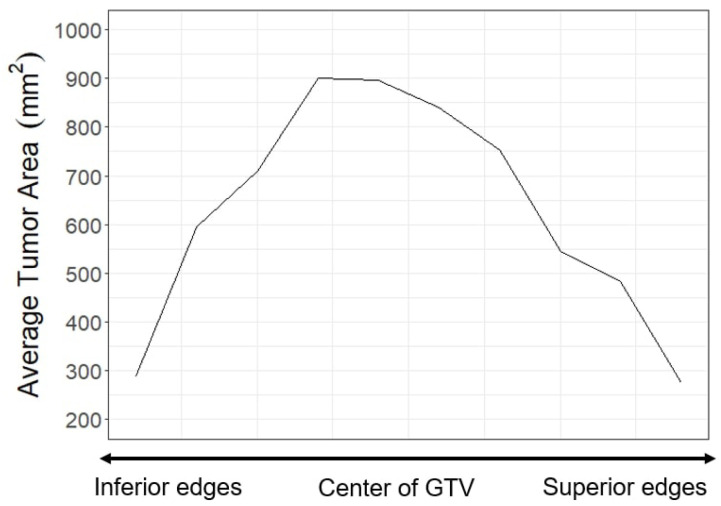
The average distribution of tumor area on a CT slice from the middle of the tumor to the superior and inferior edges. The area decreases as slices move away from the middle. These slices with low areas also correspond to slices with low performance in baseline models.

**Figure 4 diagnostics-13-02159-f004:**
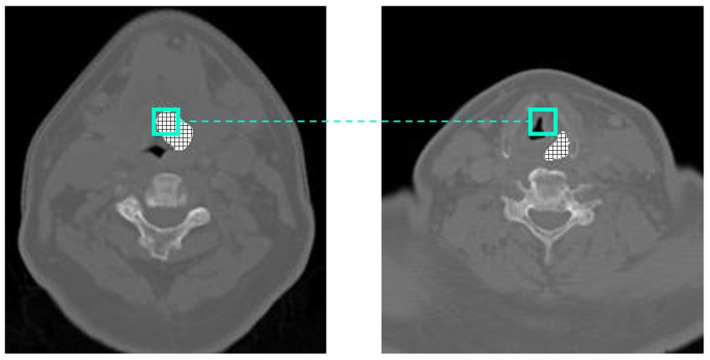
Displacement between center and edge slice of GTV.

**Figure 5 diagnostics-13-02159-f005:**
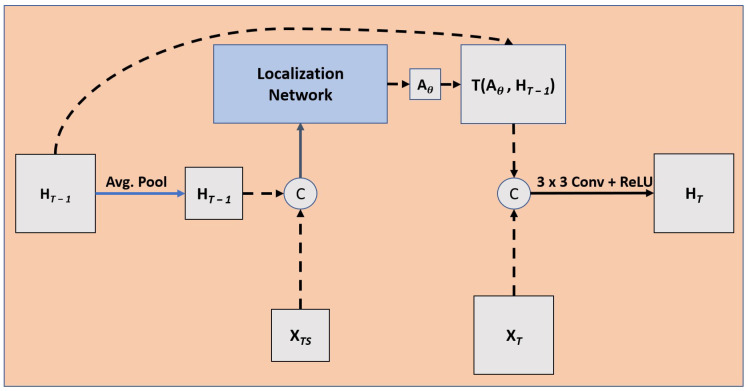
SARC block, where $X_{T}$ is the current upsampled representation of the spatial step on the spinal axis direction, $X_{T_{S}}$ is the current spatial step before the upsampling operation was applied, $H_{T}$ is the hidden state, and $A_{\theta }$ is the 6-parameter affine transformation. This shows a more detailed view of the SARC block seen in Figure 2.

**Figure 6 diagnostics-13-02159-f006:**
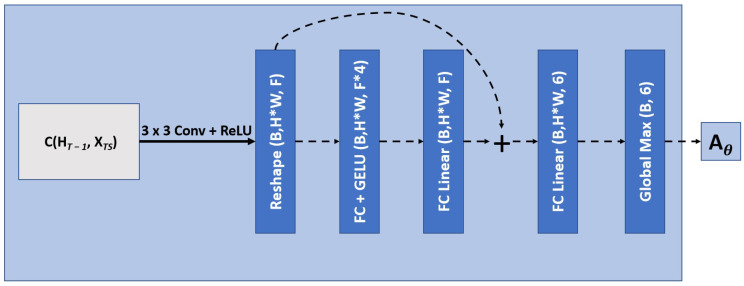
Localization network to achieve a 6-parameter affine transformation. F corresponds to the filter size, H is the image height, W is the width, and B is the batch size. FC corresponds to a fully connected layer. The filter size for the convolution is equal to the filter size for the current upsampling block. This module is a more detailed view of the localization network seen in Figure 5.

**Figure 7 diagnostics-13-02159-f007:**
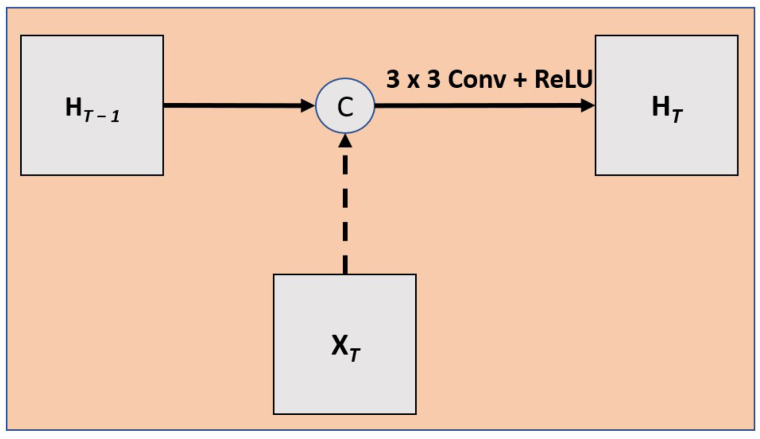
This shows the RC block, which is simpler than the SARC block, since it does not have the STN component, allowing for a spatial transformation of the hidden state. Instead, the RC U-Net only uses recurrent convolutions.

**Figure 8 diagnostics-13-02159-f008:**
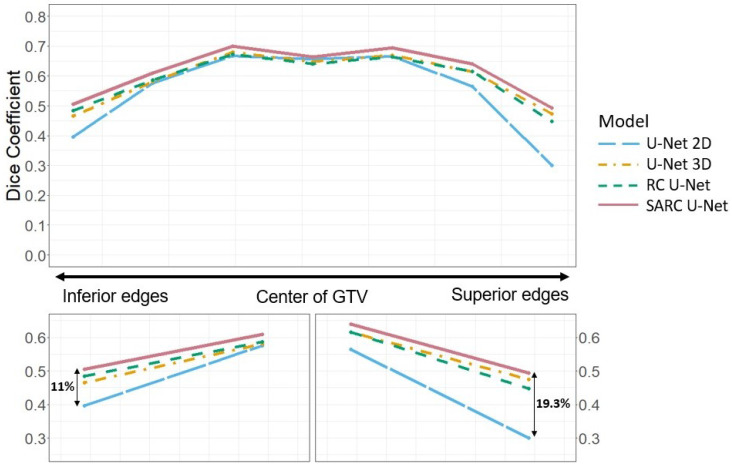
Dice coefficient by slice distance from the middle for multi-modal CT/PET inputs calculated with our base U-Net against the SARC, RC, 3D, and 2D architectures. A higher Dice coefficient means a more accurate model.

**Figure 9 diagnostics-13-02159-f009:**
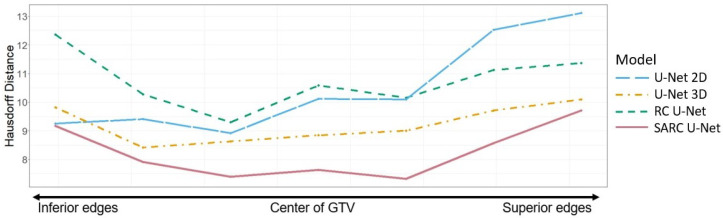
Hausdorff distance by slice distance from the middle for multi-modal CT/PET inputs calculated with our base U-Net against the SARC, RC, 3D, and 2D architectures. A lower Hausdorff distance means a more accurate model.

**Figure 10 diagnostics-13-02159-f010:**
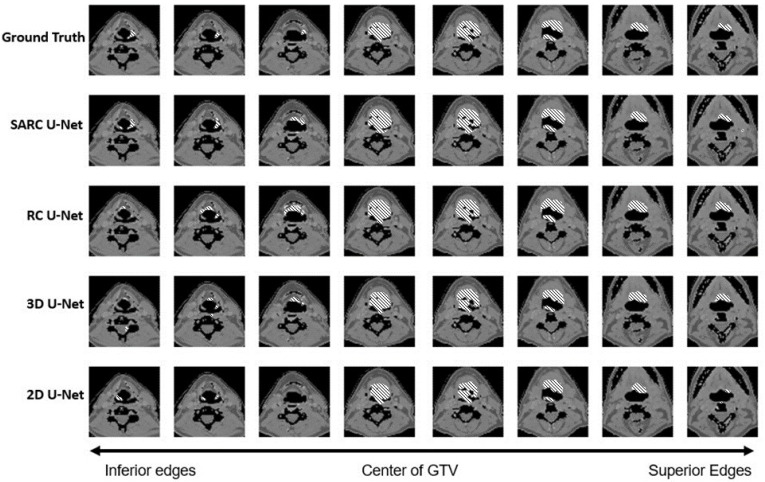
GTV contour for variations of our base model vs. ground truth. Predictions were made with both CT and PET imaging.

**Table 1 diagnostics-13-02159-t001:** Model specific information for each U-Net implementation. Inference speed is time it takes to contour a GTV for a single patient and their CT/PET scan.

Architecture	Parameters	Training Speed (s per Epoch)	Inference Speed (s per Patient)	Processes Inter-Slice Connectivity	Spatial Alignment of Anatomy in Prior Slices
2D U-Net	3,832,321	2.94	0.002	No	No
3D U-Net	10,100,353	4.26	0.002	Yes	No
RC U-Net	5,399,521	6.18	0.028	Yes	No
SARC U-Net	7,841,465	14.57	0.100	Yes	Yes

**Table 2 diagnostics-13-02159-t002:** Differences between our proposed models and previous related work.

Architecture	Processes Inter-Slice Connectivity (3D or Recurrence)	Spatial Alignment of Anatomy in Prior Slices	Begins Contouring from Center of Region of Interest (ROI)	Pushes Info. from Salient Regions of ROI to the Least Salient Regions	Applied to HNC
2D U-Net [7,16]	No	No	No	No	Yes
Spider U-Net [18]	Yes	No	No	No	No
LSTM Multi-Modal U-Net [17]	Yes	No	No	No	No
Dense 3D U-Net [8]	Yes	No	No	No	Yes
RC U-Net (ours)	Yes	No	Yes	Yes	Yes
SARC U-Net (ours)	Yes	Yes	Yes	Yes	Yes

**Table 3 diagnostics-13-02159-t003:** Cross-validated average DSC calculated over both slices and full volumes for multi-modal CT-PET inputs as well as average sensitivity and specificity over slices. Bold values indicate that the model both performed highest for a particular metric and is significantly higher (*p* < 0.05) than all other models through paired *t*-tests. The average row indicates performance averaged across all model architectures that use 2D, 3D, RC or SARC blocks.

U-Net Architecture		Mean DSC by Slice	Mean DSC by Volume	Mean Sensitivity by Slice	Mean Specificity by Slice
U-Net	2D	0.540 ± 0.078	0.561 ± 0.084	0.599 ± 0.059	0.990 ± 0.004
	3D	0.586 ± 0.068	0.547 ± 0.077	0.655 ± 0.056	0.990 ± 0.004
	RC	0.582 ± 0.074	0.549 ± 0.070	0.674 ± 0.080	0.987 ± 0.006
	SARC	**0.613 ** ± 0.075	**0.572** ± 0.067	**0.685** ± 0.064	0.990 ± 0.004
TransU-Net	2D	0.534 ± 0.069	0.540 ± 0.064	0.632 ± 0.048	0.987 ± 0.005
	3D	0.586 ± 0.057	0.548 ± 0.060	0.645 ± 0.055	0.991 ± 0.003
	RC	0.581 ± 0.066	0.541 ± 0.074	0.651 ± 0.044	0.991 ± 0.003
	SARC	**0.610** ± 0.069	**0.574** ± 0.067	**0.679** ± 0.044	0.991 ± 0.005
U-Net++	2D	0.545 ± 0.080	0.559 ± 0.069	0.628 ± 0.031	0.989 ± 0.006
	3D	0.585 ± 0.059	0.559 ± 0.071	0.628 ± 0.050	0.989 ± 0.003
	RC	0.581 ± 0.070	0.543 ± 0.066	**0.677** ± 0.031	0.989 ± 0.005
	SARC	**0.610** ± 0.073	**0.571** ± 0.068	0.665 ± 0.063	0.991 ± 0.003
Average	2D	0.540	0.553	0.620	0.989
	3D	0.586	0.551	0.643	0.990
	RC	0.581	0.544	0.667	0.989
	SARC	**0.611**	**0.572**	**0.676**	0.991

## Data Availability

For this study, data from the Cancer Imaging Archive HN1 Dataset [9,10] described in Aerts et al. [11] was used. Data can be found at https://wiki.cancerimagingarchive.net/display/Public/HEAD-NECK-RADIOMICS-HN1, (accessed on 8 August 2022) HN1 Dataset.

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
