# Peer review of "Deep Learning Architecture to Improve Edge Accuracy of Auto-Contouring for Head and Neck Radiotherapy"

_diagnostics, 2023, doi:10.3390/diagnostics13132159_

Round 1
Reviewer 1 Report
The example of the lesion shown in this manuscript has edges on one side of air cavity and on the other side that of bone is easy to contour since both densities are different in comparison to tumor tissue.
How does this model hold when the tumor's edges are not much different with surrounding soft tissue (density)?
A brief explanation of these questions will make this paper easily understood by the clinicians.
Reviewer 2 Report
The paper introduces an innovative methodology Spatially Adjusted Recurrent Convolution U-Net (SARC U-Net) for improving the edge accuracy of auto contouring for Head and Neck radiotherapy. The paper is well written, however, some points in the paper require explanation to enhance the reader's understanding.
· The authors are requested to add a section of related work to the paper. Also, include a comparison table that shows the strengths and weaknesses of the previous models and your proposed model.
· The paper compares the proposed SARC U-Net with baseline models but does not provide a direct comparison with other state-of-the-art auto-contouring methods. Including such a comparison would provide a more comprehensive evaluation of the architecture's performance.
· The paper briefly mentions that the proposed architecture achieves improved accuracy without increasing the strain on computational resources. However, it does not provide specific details or analysis regarding the computational requirements of the model. Providing information on factors such as training time, memory consumption, and inference speed would help assess the practical feasibility and scalability of the proposed approach.
· The authors are requested to provide detailed information about the experimental environment.
Round 2
Reviewer 2 Report
Most of my comments are addressed. I recommend acceptance of this article in its current form.